# Effect of Antibacterial Root Canal Sealer on Persistent Apical Periodontitis

**DOI:** 10.3390/antibiotics10060741

**Published:** 2021-06-18

**Authors:** Zheng Wang, Ge Yang, Biao Ren, Yuan Gao, Xian Peng, Mingyun Li, Hockin H.K.Xu, Qi Han, Jiyao Li, Xuedong Zhou, Lei Cheng

**Affiliations:** 1State Key Laboratory of Oral Diseases, National Clinical Research Centre for Oral Diseases, West China School of Stomatology, Sichuan University, Chengdu 610041, China; 2019324035018@stu.scu.edu.cn (Z.W.); 2016224035013@stu.scu.edu.cn (G.Y.); renbiao@scu.edu.cn (B.R.); gaoyuan@scu.edu.cn (Y.G.); pengx@scu.edu.cn (X.P.); limingyun@scu.edu.cn (M.L.); hanqi992011@163.com (Q.H.); jiyaoliscu@163.com (J.L.); 2Department of Advanced Oral Sciences and Therapeutics, Biomaterials & Tissue Engineering Division, School of Dentistry, University of Maryland, Baltimore, MD 21201, USA; hxu2@umaryland.edu

**Keywords:** persistent apical periodontitis, root canal sealer, dimethylaminododecyl methacrylate, *Enterococcus faecalis*, beagle dogs

## Abstract

The infection of *Enterococcus faecalis* and its interacting microorganisms in the root canal could cause persistent apical periodontitis (AP). Antibacterial root canal sealer has favorable prospects to inhibit biofilms. The purpose of this study was to investigated the antibacterial effect of root canal sealer containing dimethylaminododecyl methacrylate (DMADDM) on persistent AP in beagle dogs for the first time. Persistent AP was established by a two-step infection with *Enterococcus faecalis* and multi-bacteria (*Enterococcus faecalis*, *Lactobacillus acidophilus*, *Actinomyces*
*naeslundii*, *Streptococcus gordonii*). Root canal sealer containing DMADDM (0%, 1.25%, 2.5%) was used to complete root canal filling. The volume of lesions and inflammatory grade in the apical area were evaluated by cone beam computer tomography (CBCT) and hematoxylin-eosin staining. Both *Enterococcus-faecalis-* and multi-bacteria-induced persistent AP caused severe apical destruction, and there were no significant differences in pathogenicity between them. DMADDM-modified sealer significantly reduced the volume of periapical lesion and inflammatory grade compared with the control group, among them, the therapeutic effect of the 2.5% group was better than the 1.25% group. In addition, *E.*
*faecalis*-induced reinfection was more sensitive to the 2.5% group than multi-bacteria reinfection. This study shows that root canal sealer containing DMADDM had a remarkable therapeutic effect on persistent AP, especially on *E. faecalis*-induced reinfection.

## 1. Introduction

Persistent apical periodontitis (AP) has been reported to occur in 5–15% of teeth, even when the most careful clinical procedures are followed, leading to apical lesions persist after the failure of endodontic treatment [1,2]. Anatomical complexity of the root canal system and persistent infection of microorganism is the main reason for persistent AP [3,4]. Studies have shown that the most frequently detected microbe in the reinfected root canal is *Enterococcus faecalis* (24–77%) [5,6,7], because of its ability to survive in harsh environments, including extreme alkaline pH, deprivation of nutrition and antimicrobial resistance [8,9]. Recently, a study considered *Enterococcus faecalis*, *Streptococcus gordonii*, *Actinomyces naeslundii*, and *Lactobacillus acidophilus* as “core microbes” in persistent AP because of their high detection rate and the ability to form a stable biofilm structure [10]. Nevertheless, whether there are differences of pathogenicity between the monoinfection of *E.*
*faecalis* and infection caused by the four multiple species remains unresolved.

The traditional animal model for the study of apical periodontitis was induced by exposing the pulp cavity to the oral environment, to simulate the clinical pathological progression of primary root canal infection [11,12]. However, in secondary intraradicular infections post root canal therapy, with the reduction of availability resources, there exists an obvious shift in bacteria community from anaerobic Gram-negative bacteria and several Gram-positive rods into some Gram-negative bacteria, such as streptococci, *Actinomyces* species, lactobacilli and *Enterococci* [13]. At present, little research has been done on the animal study of persistent AP. In 2015, Lu et al. successfully established persistent AP models of rats with *E. faecalis* to mimic the clinical situation, which was accomplished by a two-step infection procedure [14]. However, because of the narrow pulp chamber and tiny root canal in the teeth of rats, it is subject to greater endodontic treatment failure by missing the root canal, teeth fracture or incomplete disinfection, making it a less-than-ideal model for the evaluation of dental materials in persistent AP. Due to the more anatomical similarity of teeth size and root canal thickness to people, beagle dogs were often used as animal models for evaluation of root canal disinfectants and root canal filling materials [15,16,17]. Thus, beagle dogs possess potential value to be an appropriate model for the study of endodontic material properties in persistent AP.

Eliminating persistent root canal infection can not only rely on the application of contemporary treatments such as dental operating microscope, apical microsurgery or intentional replantation, but also require the development of the antibacterial ability of root canal filling materials. Root canal sealers incorporated with antibacterial agents were proved to be a new strategy to improve the success rate of endodontic treatment [18,19]. Dimethylaminododecyl methacrylate (DMADDM) is a new quaternary ammonium monomer, which has been proved to show strong antibacterial ability when incorporated with different dental materials [20,21,22]. Our previous study indicated that DMADDM-modified root canal sealer has excellent apical sealing ability and biocompatibility, as well as effective inhibition on persistent AP-related multi-bacterial pathogens in vitro [23]. However, there is still a lack of research in vivo to evaluate the therapeutic effects of this new antibacterial modified sealer on animal models.

Therefore, in the present study, we intend to establish persistent AP models on beagle dogs infected with *E.*
*faecalis* and multi-bacteria for the first time, and investigate the therapeutic effects of DMADDM-modified root canal sealer on these two types of persistent AP.

## 2. Results

### 2.1. The Volume and Inflammatory Grade of Apical Lesions of Persistent AP

In the present study, the persistent AP models were established by a two-step infection procedure (Figure 1). Compared to the control group (non-infected canals), the periapical lesion appeared in all infection groups (Figure 2a,b). At 4 weeks after root canals exposure, the shadow volume in the apical area increased to 7.06 ± 2.65 mm^3^, indicating that the chronic AP models were successfully established. After root canal preparation and disinfection, no significant differences of periapical lesion volume were observed compared to the initial infection group (*p* > 0.05). Therefore, we speculated that the treatment had an antibacterial effect to stop the periapical bone destruction. Through inoculation with *E.*
*faecalis* and multi-bacteria, the shadow volume in apical increased by 9.15 ± 2.63 mm^3^ and 9.67 ± 2.23 mm^3^, respectively, compared to the initial infection group (*p* < 0.01). This result shows that the periapical destruction was aggravated, indicating that both single- and multi-bacteria-induced persistent AP models were established. Interestingly, we found there were no differences in periapical destruction between *E.*
*faecalis* and the multi-bacteria group (*p* > 0.05).

For histopathological examination (Figure 2c and Table 1), there was no apical bone destruction and inflammatory cell infiltration in the control group. In contrast to this, the inflammatory cells in the initial infection group increased significantly (*p* < 0.01) and infiltrated beyond the apical foramen with moderate apical bone resorption. After disinfection, the number of inflammatory cells decreased significantly (*p* < 0.01), accompanied with obvious bone hyperplasia. After bacteria re-infection, inflammation entered the acute phase, a more serious inflammatory cell infiltration happened, and localized bleeding occurred, as well as severe apical bone and root resorption, without any bone hyperplasia. There were no differences in periapical inflammation grade between *E.*
*faecalis* group and multi-bacteria group.

### 2.2. Analysis of the Antibacterial Effect of DMADDM-Modified Sealer on Persistent AP

In order to evaluate the therapeutic effect of DMADDM-modified sealer on the two different types of persistent AP, we used EndoREZ with a different mass fraction (0%, 1.25%, 2.5%) of DMADDM to complete root canal filling. The results reveal that in *E.*
*faecalis* and multi-bacteria-induced persistent AP, the apical shadow volume of the 0% DMADDM group continued to expand by 4.76 ± 1.83 and 5.77 ± 2.45 mm^3^, respectively (Figure 3a). Conversely, the sealers containing 1.25% and 2.5% DMADDM significantly reduced the volume of periapical lesion (*p* < 0.01). Among them, the reduction of apical lesions in 2.5% group was more obvious than that in 1.25% group (*p* < 0.01). When comparing the two different types of persistent AP, there were no significant differences in the apical lesion reduction of the 1.25% group. However, the 2.5% group can reduce more apical lesion in *E.*
*faecalis*-induced persistent AP than the multi-bacteria group.

The histopathology indicated that in both *E.*
*faecalis*-induced persistent AP and multi-bacteria-induced persistent AP, the 0% group showed a greater degree of inflammation than that before the root canal filling (Figure 3b and Table 2). In the 1.25% group, there was local infiltration with chronic inflammatory cells such as neutrophils and lymphocytes; in addition, neovascularization was observed at the apical foramen, indicating that inflammation entered the chronic phase. The 2.5% group contained neoformed cementoid tissue and some neovascularization in the apical area, and the number of inflammatory cells was significantly reduced (*p* < 0.01), which showed that periapical lesions were being repaired.

### 2.3. Morphology and Proliferation Activity of Biofilm Models In Vitro

From the CBCT results, we found that multi-bacteria-induced persistent AP was more resistant to 2.5% DMADDM-modified sealer compared with the *E.*
*faecalis* group. In order to validate whether *E.*
*faecalis* is more virulent when coexistent with the other three bacteria, biofilm models were established in vitro for the phenotypic study of *E.*
*faecalis*. The scanning electron microscope (SEM) images showed that there were a large number of *E. faecalis* cells within the single-species biofilm, accompanied with a few extracellular matrices (Figure 4a). In contrast, multi-bacterial biofilm was significantly thicker and denser than that formed by *E.*
*faecalis*, and a mass of EPS was distributed among cells, by which *E.*
*faecalis* adhered to other species. For live/dead imaging, in both *E.*
*faecalis* and multispecies biofilm, the red staining was equal to the green staining, indicating that the proliferation of *E.*
*faecalis* cells was similar to multispecies cells (Figure 4b,c). The real-time polymerase chain reaction results reveal that there were no significant differences in the number of *E.*
*faecalis* between single and multispecies biofilm at 24 h (Figure 4d). However, multispecies biofilm had much more *E.*
*faecalis* at 48 h (*p* < 0.01).

## 3. Discussion

In the present study, we established persistent AP animal models on beagle dogs for the first time and investigated the antibacterial effect of DMADDM-modified root canal sealer. Persistent AP models were constructed based on a previous study [14], which was accomplished by a two-step infection procedure. Compared with the previous study, our research has made some progress. Firstly, although rat models have existed in some AP studies [24,25], it is a challenge to test the properties of materials because of the extremely narrow root canal system and the treatment complications. Thus, problems due to the tiny teeth were overcome by using beagle dog models in the present study. Our research has shown that persistent AP could be produced successfully in beagle dog models, which were confirmed by CBCT and HE stains. Secondly, the previous study [9] has a limitation with the monospecies infection of *E. faecalis*. It is well known that monoinfections seldom occur in nature [26,27], and some researchers have shown that persistent AP is primarily caused by multispecies bacteria [28]. Therefore, in this experiment, it has been modified to establish persistent AP models infected with *E. faecalis* and multi-bacteria. Our results indicated that there were no significant differences in pathogenicity between the two types of persistent AP, probably because bacteria could grow in biofilms or coaggregation forms that adhere to dentinal tubules or other regions of the root canal, thereby damaging the periapical tissue [3].

The development of persistent AP is a long process in which the root canal system evolves from communicating with the oral cavity into an enclosed environment [5]. In the present study, chemomechanical preparation and intracanal disinfection were performed to remove pathogenic bacteria in primary chronic apical periodontitis. Calcium hydroxide as the disinfection drug commonly used in clinic treatment can promote the inflammation absorption and bone tissue repair [29]. However, its antibacterial effect on *E. faecalis* did not work well [30]; thus, in this experiment, calcium hydroxide was used to kill and inhibit common pathogenic bacteria after initial infection to ensure that exogenous inoculation plays the dominant role in the re-infection procedure.

DMADDM was synthesized and demonstrated to be a strong antibacterial agent when incorporated with root canal sealer. In the present study, EndoREZ with different mass fractions of DMADDM exerted significant antibacterial properties. The CBCT and pathologic results indicate that sealer containing DMADDM significantly decreased the apical lesion volume and inflammation degree. Moreover, increasing the concentration of DMADDM achieved a stronger antibacterial effect. It widely recognized that quaternary ammonium salts (QAS) has a “contact killing” effect on oral bacteria due to their capability of causing bacteria lysis by binding to bacterial membranes [31,32]. As a flowable filling material, EndoREZ can flow into accessory canals and dentinal tubules to facilitate resin tag formation for retention and sealing after smear layer removal [33]. Therefore, in present study, EndoREZ modified with DMADDM could kill the bacteria colonized in the root canal system, such as in isthmuses, dentinal tubules and ramifications, directly to achieve the desired effect of root canal disinfection. In addition, our previous study indicated that DMADDM may indirectly inhibit bacteria without contact with them by influencing the competitions among bacteria during the growth of biofilms [34].

Interestingly, the response of *E. faecalis*-induced persistent AP was more sensitive to 2.5% DMADDM than multi-bacteria. We hypothesize that this result might be caused by the differences in biofilm formation ability between the single *E. faecalis* strain and multi-bacteria. Our results in vitro reveal that *E. faecalis* grew more remarkably after 48 h when cultured in multispecies biofilm. Additionally, the SEM images indicate that the extracellular matrix in multispecies biofilm was significantly more than that of the *E. faecalis* alone, which may be produced by the other three bacteria and enable *E. faecalis* to become more firmly adhered to the biofilm. If we combine the results of RT-PCR and SEM, the enhanced growth activity and protective effects of the extracellular matrix could enable multi-bacterial infection more resistant to DMADDM-modified sealer. Similar consequences were found in a previous study, namely, *E. faecalis* was more resistant to starvation in coexistence with other pathogenic species [26]. When there is sufficient nutrition in the environment, *E. faecalis* can keep a dynamic balance with other bacteria because of the complex interaction between multispecies. However, under starvation conditions such as root-filled canals, *E. faecalis* can produce bacteriocins, including enterocin and cytolysin, to inhibit other bacteria in interspecies competition [35,36]. For example, Chavez de Paz found that *E. faecalis* can inhibit *A. naeslundii* in the dual-species model [37]. These interspecific interactions might be a reason why *E. faecalis* generates overproliferation in a multispecies model, leading to an enhanced resistance to DMADDM after root canal therapy.

This study also has some limitations. It is remarkable that even though the apical shadow volumes of the 1.25% and 2.5% DMADDM groups were significantly reduced compared to the control group, they were not reversed to the level of chronic apical periodontitis. This probably means that lesions in most teeth were relatively large, and three months is insufficient to completely repair apical bone destruction. It also reminds us of the importance of enhancing the combination of antibacterial modification sealer with apical microsurgery and intentional replantation in some persistent cases.

## 4. Materials and Methods

### 4.1. Bacterial Strains and Culture Conditions

The bacterial-contained *Enterococcus faecalis* ATCC29212, *Streptococcus gordonii* ATCC35105, *Actinomyces naeslundii* ATCC12104 and *Lactobacillus acidophilus* ATCC4356 were provided by the State Key Laboratory of Oral Diseases, Sichuan University. Bacteria were incubated at 37 °C (90% N_2_, 5% CO_2_, 5% H_2_) in a brain–heart infusion broth (BHI; Difco, Sparks, MD, USA).

### 4.2. Synthesis of DMADDM and Modified Root Canal Sealer Preparation

DMADDM was synthesized and verified in a previous study [38] (Appendix A). DMADDM was mixed with EndoREZ (Ultradent, South Jordan, UT, USA), at a DMADDM/(EndoREZ + DMADDM) mass fraction of 0% (control group), 1.25%, and 2.5%, agitated for 5 min and light-cured for 10s to promote both chemical structures of the main components in DMADDM and EndoREZ feature double bonds form cross-link structures (Appendix A).

### 4.3. Animal Preparations

Seven beagle dogs were obtained at the age of 12 months (weight 8–12 kg) in this work. In total, 64 root canals from the upper and lower premolars were utilized.

### 4.4. Establishment of Persistent AP Models

Four beagle dogs were anaesthetized by intraperitoneal injection with 3% pentobarbital sodium (30 mg/kg body weight, Aikonchem, Jiangsu, China) and the pulp chambers of the teeth studied were opened. The canals were exposed to the oral environment for 4 weeks to induce a chronic AP. Cone beam computed tomography (CBCT) was taken to observe the periapical lesion. One of the dogs was sacrificed, and its canals were randomly divided into two groups, including 10 exposed canals and 10 non-infected canals (control group), for hematoxylin and eosin (H&E) stains. Periradicular tissue responses were histologically classified into four categories, according to the amount of inflammatory cells and the degree of apical bone and root destruction (Appendix A).

Four weeks later, the canals of another 3 dogs were instrumented using K-files and finished at #25 by the step-back technique, with irrigation by 1.0% sodium hypochlorite (Longly, Wuhan, China). Then, the root canals were dried with sterile paper points and filled with calcium hydroxide (Ivoclar Vivadent, Schaan, Liechtenstein), followed by access cavity sealing with Filtek^TM^Z350XT adhesive resin (3M, St. Paul, MN, USA). CBCT was taken to observe the periapical lesion for all 3 dogs, and one of the dogs was sacrificed for histopathological examination.

After 2 weeks of disinfection, the canals of the remaining two dogs were divided into the control group, *E. faecalis* group and multi-bacteria group. Then, the canals were prepared to #30 and irrigated with 1.0% sodium hypochlorite (Longly, Wuhan, China), followed by inoculation of bacterial suspension. For the *E.*
*faecalis* group, the bacteria concentration was 10^6^ colony-forming units [CFUs]/mL. For the multi-bacteria group, bacterial suspensions were mixed to obtain an inoculum containing *E.*
*faecalis* (10^6^ CFUs/mL), *L. acidophilus* (10^6^ CFUs/mL), *A. neisseriae* (10^6^ CFUs/mL) and *S. gordonii* (10^6^ CFUs/mL). For the control group, the root canals were injected with sterile normal saline. The cavity was sealed with adhesive resin (3M, St. Paul, MN, USA). Two weeks later, CBCT was taken to observe the periapical lesion for 2 dogs, and all of them were sacrificed for histopathological examination.

### 4.5. Antibacterial Efficiency of Modified Sealer on Persistent AP

The persistent AP models were established as shown above. The root canals of the remaining three beagle dogs were divided into 6 groups, half of which were re-infected with *E.*
*faecalis*, and the other half were re-infected with multi-bacteria. After removing the filling contents in cavity, root canals were prepared to #35, and irrigated with 1.0% sodium hypochlorite (Longly, Wuhan, China). After being dried with paper points, the root canals were filled with gutta-percha points and DMADDM (0%, 1.25%, 2.5%) modified sealer by the lateral condensation technique, followed by access cavity sealing with adhesive resin (3M, St. Paul, MN, USA).Three months after treatment, CBCT was taken to observe the periapical lesion. Images were performed with MIMICS V21.0 to calculate the shadow volume change in the apical area (Appendix A). All three dogs were sacrificed for histopathological examination.

### 4.6. Biofilm Formation and Imaging

Single-species and multispecies biofilm were prepared following a previous study [23]. For single-species biofilm formation, *E. faecalis* cells were harvested at mid-logarithmic phase and incubated in 2 mL of BHI with 1% sucrose (10^6^ colony-forming units (CFUs)/mL). For multispecies biofilm formation, bacterial suspensions mixed in BHI with 1% sucrose were cultured in 24-well plates which composed of the four species: *E. faecalis* (10^6^ CFUs/mL), *S. gordonii* (10^6^ CFUs/mL), *A. naeslundii* (10^6^ CFUs/mL), and *L. acidophilus* (10^6^ CFUs/mL). The culture medium refreshed every 24 h.

The biofilm imaging included scanning electron microscopy (SEM) and live/dead bacteria staining, which were studied as previously described [39,40,41]. Scanning electron microscopy (SEM) imaging was used to observe the morphology of bacteria and the extracellular matrix, and biofilm samples at 48 h were rinsed with PBS and immersed in 2.5% glutaraldehyde for 6 h. Then, the biofilm was serially dehydrated with ethanol (50%, 60%, 70%, 80%, 90%, 95%, and 100%), sputter-coated with gold and examined by scanning electronic microscopy (FEI, Hillsboro, OR, USA). For live/dead bacteria staining, the bacterial cells in biofilms at 48h were labeled with 2.5 μM SYTO9 (Molecular Probes, Invitrogen) and propidium iodide (Molecular Probes) for 15 min, and the images were captured with a confocal laser scanning microscope (TCS SP8, Leica, Wetzlar, Germany) equiped with 60× oil immersion objective lens.

### 4.7. DNA Isolation and Real-Time Polymerase Chain Reaction

The amount of bacteria was investigated with a technique described by Zhang [34]. Single-species and multispecies biofilms were harvested as shown above. Total bacterial DNA of the 24 h and 48 h biofilms were isolated by a TIANamp Bacteria DNA kit (TIANGEN, Beijing, China). Additionally, the purity and concentration tests for DNA were performed by a NanoDrop 2000 spectrophotometer (Thermo Scientific, Waltham, MA, USA). The amount of the DNA was performed by Quantitative Real-time PCR on a C1000 Touch™ Thermal Cycler instrument (Bio-Rad, Philadelphia, PA, USA), which was used to calculate the number of *E.*
*faecalis*, *S.*
*gordonii*, *A.*
*naeslundii*, and *L*. *acidophilus*. The sequences of the primers for the four species are listed in Appendix A.

### 4.8. Statistical Analysis

The SPSS16.0 software (SPSS Inc., Chicago, IL, USA) was used for the statistical analysis. Nonparametric Kruskal–Wallis analysis and the Mann–Whitney U-test were used to analyze the change of apical shadow volume and histopathologic results. For the in vitro studies, one-way analysis of variance and the Student–Newman–Keuls test were performed to detect the significant effects of the variables. Significant differences were considered when *p* < 0.05.

## 5. Conclusions

In conclusion, we established two different types of persistent AP models in vivo and investigated the antibacterial effect of root canal sealer containing a different mass fraction of DMADDM for the first time. The beagle dog models for persistent AP were constructed by a two-step infection procedure. Our research has shown that persistent AP could be produced successfully in beagle dog models; meanwhile, it is suitable for evaluating the properties of dental materials compared with rat models in previous studies. Moreover, the therapeutic effect of DMADDM-modified sealer on persistent AP models provides important support for clinical application.

## Figures and Tables

**Figure 1 antibiotics-10-00741-f001:**
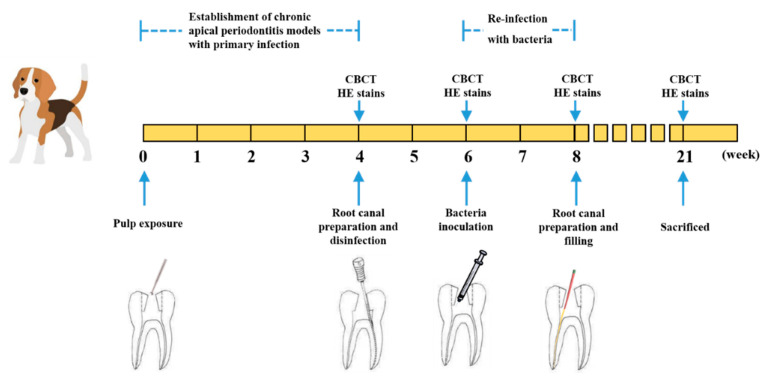
Establishment of *E.*
*faecalis* and multi-bacteria-induced persistent AP in beagle dogs and filled with DMADDM-modified sealer.

**Figure 2 antibiotics-10-00741-f002:**
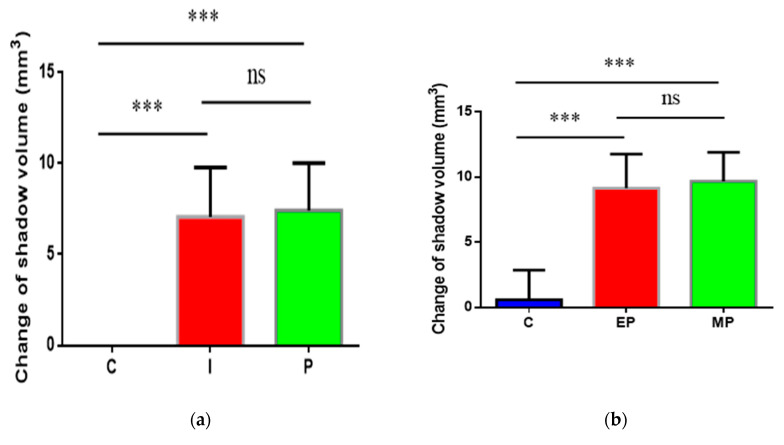
The volume and histopathological section of periapical lesions in each time point. (**a**) “C” represents the control group (non-infected canals), “I” represents the initial chronic AP group and “P” represents the root canal preparation and disinfection group; (**b**) “C” represents the control group (injected with sterile normal saline), “EP” represents *E.*
*faecalis*-induced persistent AP and “MP” represents multi-bacteria-induced persistent AP; (**c**) histopathological images in each time points. Apical periodontal ligament (APL), areas of apical resorption (triangle), and chronic inflammatory reaction (INF) in the apical periodontal space. The apical shadow volume was tested through CBCT and calculated by MIMICS V21.0. Each value is mean ± SD (*n* = 6) (*** *p* < 0.001).

**Figure 3 antibiotics-10-00741-f003:**
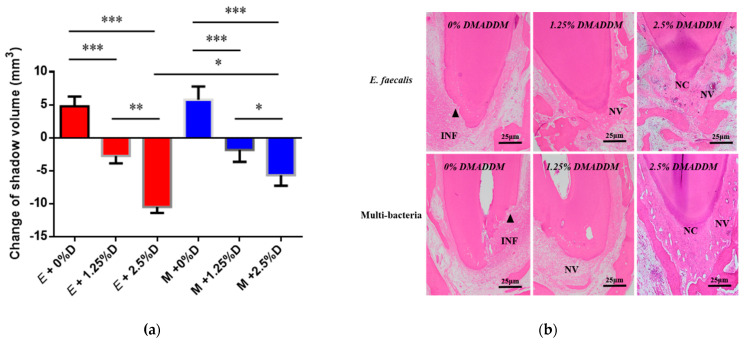
The volume and histopathological section of periapical lesions treated by DMADDM. (**a**) Change of apical shadow volume after treatment with DMADDM. “E” represents *E.*
*faecalis*, “M” represents multi-bacteria, “D” represents DMADDM; (**b**) histopathological images after treatment with DMADDM. Chronic inflammatory reaction (INF) in the apical periodontal space, neovascularization (NV) around areas of apical resorption, and neoformed cementum (NC) partially repaired areas of apical resorption. The apical shadow volume was tested through CBCT and calculated by MIMICS V21.0. Each value is mean ± SD (*n* = 6) (* *p* < 0.05, ** *p* < 0.01, *** *p* < 0.001).

**Figure 4 antibiotics-10-00741-f004:**
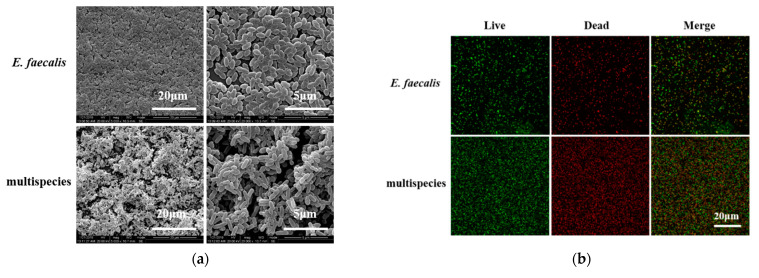
The biofilm formation of *E. faecalis* and multispecies models. (**a**) The scanning electron microscopy images of *E. faecalis* and multispecies biofilms with 5000× and 20,000× magnified visual fields; (**b**) the live/dead bacterial staining image of biofilms (live bacteria, stained green; dead cells, stained red); (**c**) the ratio of the live bacteria cells to dead cells was computed in line with 3 random sights of biofilms; (**d**) colony-forming units of bacteria in biofilms formed by *E. faecalis* and multispecies (*n* = 3). Data are presented as the mean ± standard deviation (** *p* < 0.01).

**Table 1 antibiotics-10-00741-t001:** The histopathologic results in each time points.

Groups	The Categories of Inflammation	The Average Number of Inflammatory Cells
Initial chronic AP	3	196
After root canal disinfection for 2 weeks	1	42
*E.**faecalis*-induced persistent AP	3	210
Multi-bacteria-induced persistent AP	3	193

**Table 2 antibiotics-10-00741-t002:** The histopathologic results after treatment with DMADDM.

Groups	The Categories of Inflammation	The Average Number of Inflammatory Cells
*E.**faecalis* + 0% DMADDM	3	207
*E.**faecalis* + 1.25% DMADDM	2	88
*E.**faecalis* + 2.5% DMADDM	1	30
Multi-bacteria + 0% DMADDM	3	201
Multi-bacteria + 1.25% DMADDM	2	89
Multi-bacteria + 2.5% DMADDM	1	35

## Data Availability

Data are contained within the article.

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
