# Peer review of "Effect of Antibacterial Root Canal Sealer on Persistent Apical Periodontitis"

_antibiotics, 2021, doi:10.3390/antibiotics10060741_

Round 1
Reviewer 1 Report
The article presents the results of an original in vivo study, of a preliminary nature and with a small sample.
The article needs an extensive language review .
1. Abstract:
I think when authors talk about Actinomyces Neisseriae they pretend to say Actinomyces naeslundii
2. Introduction:
Please clarify the sentence: There exist an obvious shift in bacteria community from anaerobic gram-negative bacteria and several gram-positive rods into some gram-negative bacteria ?
3. In the version sent, the various sections of the article appear in an incorrect order.
M&M
4. The study was submitted to a IRB for research on animals?
5. The number of canals treated in each dog is not clear (from a total of 54). Figure 1 is not clear enough to understand the number of dogs allocated to each group. Has any type of randomization been carried out?
6. It is only realized through a figure caption (line 112) that the canal in the control group were injected with a saline solution. Every part of the materials and methods could be better described. The way it is organized makes it difficult to read and understand study protocol.
7. Was any medication given other than anesthesia?
8. Discussion
The text in line 191 to 194 is equal to the text in line 203-206
The limitations of the study are not discussed.
No consideration is given to whether the results can be extrapolated.
Author Response
Response to Reviewer 1 Comments
The article presents the results of an original in vivo study, of a preliminary nature and with a small sample.
The article needs an extensive language review.
Point 1: Abstract: I think when authors talk about Actinomyces Neisseriae they pretend to say Actinomyces naeslundii
Response 1: Thank you very much. We revised the word “Actinomyces Neisseriae” in the new version as suggested .
Point 2: Introduction: Please clarify the sentence: There exist an obvious shift in bacteria community from anaerobic gram-negative bacteria and several gram-positive rods into some gram-negative bacteria ?
Response 2: Thank you very much. In a primary root canal infection, the open root canal provides an available resource for microbes through direct communication with the oral cavity. Anaerobic gram-negative bacteria and several gram-positive rods are often found in the root canal. However, in root-filled canals with persistent periapical disease, an obvious decrease in the availability of resources in the root canal occurs because chemomechanical preparation and intracanal medication are introduced and the root canals are sealed. Gram-negative bacteria gradually decrease, and gram-positive bacteria are more frequently present, such as streptococci, Actinomyces species, lactobacilli and Enterococci. We also revised the sentence in the new version.
Point 3: In the version sent, the various sections of the article appear in an incorrect order. M&M
Response 3: Thank you very much. We revised the order in Materials and Methods section as suggested.
Point 4: The study was submitted to a IRB for research on animals?
Response 4: Thank you very much. This study for research on animals was conducted in accordance with the Declaration of Helsinki, the policy of Sichuan University and West China School of Stomatology, and the protocol was approved by the Ethical Committee of West China School of Stomatology, Sichuan University (Chengdu, China) (Project identification code: WCHSIRB-D-2017-114, approval date: 05/25/2017).We revised the description in “Institutional Review Board Statement” section.
Point 5: The number of canals treated in each dog is not clear (from a total of 54). Figure 1 is not clear enough to understand the number of dogs allocated to each group. Has any type of randomization been carried out?
Response 5: Thank you very much. In present study, we used a simple randomization method, 7 beagle dogs were randomly numbered as 1, 2, 3, 4, 5, 6, 7. Among the canals of the No. 1, ten root canals were randomly selected for inducing of chronic AP and ten root canals were randomly selected from the remaining untreated root canals as negative controls. Eight root canals were randomly selected from the No. 2 for pulp exposure, root canal preparation and disinfection. Six root canals were randomly selected from No. 3 for pulp exposure, root canal preparation, disinfection and inoculation of E. faecalis. Six root canals were randomly selected from No. 4 for pulp exposure, root canal preparation, disinfection and inoculation of multi bacteria. Eight root canals were randomly selected from samples No. 5, 6, and 7, a total of 24 root canals, which were randomly divided into 6 groups for DMADDM antibacterial efficiency investigation. It is worth noting that the 54 root canals in this experiment refer to the processed root canals, excluding the untreated control group in the No. 1 sample (We have revised the number of root canals involved in present study). We regret that due to the complexity of the grouping, it is difficult to show the number of dogs allocated to each group on Figure 1.
Point 6: It is only realized through a figure caption (line 112) that the canal in the control group were injected with a saline solution. Every part of the materials and methods could be better described. The way it is organized makes it difficult to read and understand study protocol.
Response 6: Thank you very much. We revised the description of the Materials and Methods in the new version as suggested.
Point 7: Was any medication given other than anesthesia?
Response 7: Thank you very much. In present study, in addition to the use of anesthetics, we only used calcium hydroxide for root canals disinfection, and no other medication were given to the animals.
Point 8: Discussion
The text in line 191 to 194 is equal to the text in line 203-206
The limitations of the study are not discussed.
No consideration is given to whether the results can be extrapolated.
Response 8: Thank you very much. We deleted the duplicate content in Line 203-206 as suggested, and we expanded the discussion of the limitations of this study in the new version. The results of animal experiment in present study and in vitro experiments from our previous study (doi: 10.1038/s41598-019-47032-8) have proved that the antibacterial modified root canal sealer can indeed inhibit periapical periodontitis and its related pathogen, indicating that it may have the potential for clinical application in the future, but more experiments are needed to confirm its therapeutic effect, such as clinical trials.
Reviewer 2 Report
The authors showed the promising antibacterial activity of root canal sealer composed by dimethylaminododecyl methacrylate. Additionally, they provided a useful in vivo model, which significantly contribute to translational research. I recommend the publication of the manuscript, with a minor revision.
In Figure 2A and B the authors used three asterisks, however the figure legend only describes one asterisk. In Figure 4 the authors used two asterisks and describe the meaning of using one in the legend. Please review this.
The description of the meaning of the two and three asterisks in the legend of figure 3 is missing.
Author Response
Response to Reviewer 2 Comments
The authors showed the promising antibacterial activity of root canal sealer composed by dimethylaminododecyl methacrylate. Additionally, they provided a useful in vivo model, which significantly contribute to translational research. I recommend the publication of the manuscript, with a minor revision.
Point 1: In Figure 2A and B the authors used three asterisks, however the figure legend only describes one asterisk. In Figure 4 the authors used two asterisks and describe the meaning of using one in the legend. Please review this.
Response 1: Thank you very much. We revised the description of Figure Legend in Figure 2 and Figure 4 as suggested.
Point 2: The description of the meaning of the two and three asterisks in the legend of figure 3 is missing.
Response 2: Thank you very much. We revised the description of Figure Legend in Figure 3 as suggested.
Round 2
Reviewer 1 Report
In general, the authors clarified questions that had been raised.
Helsinky declaration applies for medical research involving human subjects.